

# Long-range RNA structures in the human transcriptome beyond evolutionarily conserved regions

Sergey Margasyuk[1], Lev Zavileyskiy[1], Changchang Cao[2] and Dmitri Pervouchine[1]

[1] Center for Molecular and Cellular Biology, Skolkovo Institute of Science and Technology, Moscow, Russia
[2] Key Laboratory of RNA Biology, Institute of Biophysics, Chinese Academy of Sciences, Beijing, China

## ABSTRACT

RNA structure has been increasingly recognized as a critical player in the biogenesis and turnover of many transcripts classes. In eukaryotes, the prediction of RNA structure by thermodynamic modeling meets fundamental limitations due to the large sizes and complex, discontinuous organization of eukaryotic genes. Signatures of functional RNA structures can be found by detecting compensatory substitutions in homologous sequences, but a comparative approach is applicable only within conserved sequence blocks. Here, we developed a computational pipeline called PHRIC, which is not limited to conserved regions and relies on RNA contacts derived from RNA *in situ* conformation sequencing (RIC-seq) experiments. It extracts pairs of short RNA fragments surrounded by nested clusters of RNA contacts and predicts long, nearly perfect complementary base pairings formed between these fragments. In application to a panel of RIC-seq experiments in seven human cell lines, PHRIC predicted ~12,000 stable long-range RNA structures with equilibrium free energy below −15 kcal/mol, the vast majority of which fall outside of regions annotated as conserved among vertebrates. These structures, nevertheless, show some level of sequence conservation and remarkable compensatory substitution patterns in other clades. Furthermore, we found that introns have a higher propensity to form stable long-range RNA structures between each other, and moreover that RNA structures tend to concentrate within the same intron rather than connect adjacent introns. These results for the first time extend the application of proximity ligation assays to RNA structure prediction beyond conserved regions.

## INTRODUCTION

RNA structure plays a critical role in the maturation of eukaryotic transcripts (*Baralle, Singh & Stamm, 2019*; *Jacobs, Mills & Janitz, 2012*). Several lines of evidence indicate that it is largely involved in determining the outcome of pre-mRNA splicing at multiple levels, from assisting the spliceosome to choose the correct splice sites to modulating backsplicing events that give rise to circular RNAs (*Warf & Berglund, 2010*). Double-stranded regions

Corresponding author
Dmitri Pervouchine,
D.Pervouchine@skoltech.ru

in mammalian genes have been increasingly reported as implicated in human disease and serve as targets for small-molecule drugs and antisense oligonucleotides (*Singh, Singh & Androphy, 2007*; *Garcia-Lopez et al., 2018*).

Evolutionary conservation and independent compensatory phylogenetic substitutions are typical features that distinguish functional RNA structures from random base pairings (*Rivas, 2021*). Conserved RNA elements identified by phylo-HMM (*Felsenstein & Churchill, 1996*; *Blanchette et al., 2004*) allowed characterization of distinct properties of conserved complementary regions in human introns, many of which possess evolutionary signatures (*Kalmykova et al., 2021*). However, measurements of nucleotide covariations in multiple sequence alignments are impossible outside of conserved blocks, which limits the scope of phylogenetic RNA structure prediction to exons and conserved intronic RNA elements. Exons, on the other hand, evolve under the evolutionary constraint of maintaining the protein aminoacid sequence, which confounds the phylogenetic signatures that are characteristic for RNA base pairings. Considering that conserved RNA elements constitute only 5% of the human intronic sequences, functional RNA structures beyond evolutionarily conserved regions remain largely unknown.

The development of high throughput sequencing methods enabled the assessment of RNA structure by a number of strategies including proximity ligation assays (*Xu et al., 2022*; *Wang et al., 2021*). These experiments use digestion and stochastic religation of crosslinked RNA molecules to assess their spatial proximity, thus offering a snapshot of not only hairpin-like but also long-range RNA structures, which are formed over large distances in the primary nucleotide sequence (*Lu et al., 2016*; *Sharma et al., 2016*; *Aw et al., 2016*; *Ziv et al., 2018*). Among them, of special interest is the novel class of proximity ligation assays called RNA *in situ* conformation sequencing (RIC-seq), in which RNAs are crosslinked through RNA-binding proteins, thus enabling the assessment of RNA structure formed within physiological cellular complexes (*Cai et al., 2020*; *Cao et al., 2021*). We showed recently that RIC-seq strongly supports conserved vertebrate RNA structures that were identified bioinformatically, particularly those with unique features such as equilibrium free energy, the occurrence of A-to-I RNA editing sites, and forked eCLIP peaks (*Margasyuk et al., 2023a*).

Here, we approach the identification of long-range RNA structures assuming that they are surrounded by RNA contacts observed in proximity ligation experiments (Fig. 1). Towards the identification of such structures, we developed a computational pipeline called PHRIC, which identifies pairs of nested contact clusters (PNCC, see below) using RIC-seq data, extracts nucleotide sequences enclosed between contacts, and performs thermodynamic RNA folding of the extracted sequences to identify long, nearly perfect complementary matches. We applied it to a panel of RIC-seq experiments in seven human cell lines including GM12878, H1, HeLa, HepG2, IMR90, K562, and hNPC to characterize long-range RNA interactions in exons and introns of human genes without taking into account sequence conservation. We predicted a core set of 11,998 RNA structures including exonic, intronic, and mixed RNA structures, the majority of which were located

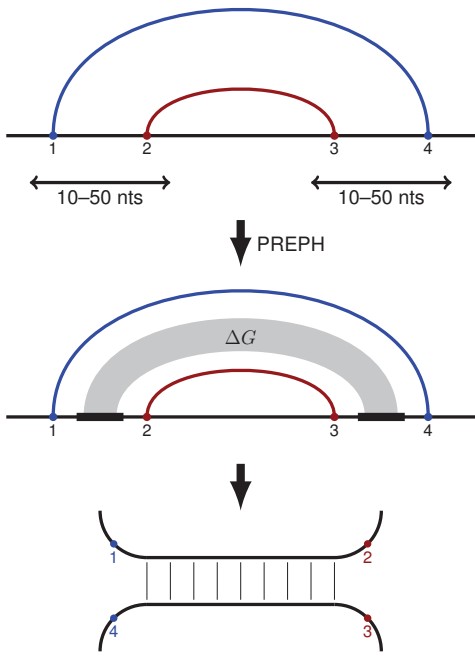

**Figure 1 PHRIC pipeline.** RNA contacts from RIC-seq data are clustered, and pairs of nested contact clusters (PNCC) are identified (top). The nucleotide sequences located between the 5′-ends (1 and 2) and the 3′-ends (3 and 4) of the inner and the outer contact are extracted and passed to PREPH (prediction of panhandles) software to detect long, almost perfect complementary regions (middle). The RNA structure formed by these regions (bottom) is supported by RNA contacts (1–4) and (2–3) identified by the proximity ligation assay.

outside of conserved regions and not characterized previously. Unexpectedly, we found that introns have a higher propensity to form stable RNA structures between themselves as compared to exons despite higher GC content and hence larger contribution of stacking energies to the RNA structure stability from exonic sequences. In conclusion, we discuss a number of remarkable RNA structures with potential impact on splicing that were predicted by PHRIC pipeline and visualize them through Genome Browser tracks (*Raney et al., 2014*).

## METHODS

### RIC-seq and RNA-seq experiments

The results of RIC-seq experiments conducted in seven human cell lines including GM12878, H1, HeLa, HepG2, IMR90, K562, and hNPC (two bioreplicates each) were downloaded from the Gene Expression Omnibus under the accession numbers GSE127188 and GSE190214 in FASTQ format. The matched control RNA-seq experiments were downloaded from the ENCODE consortium under the accession numbers listed in Table S1. RIC-seq and the matched RNA-seq data were processed by `RNAcontacts` pipeline (*Margasyuk et al., 2023d*) using February 2009 (hg19) assembly of the human genome and GENCODE transcript annotation v41lift37, which were downloaded from Genome Reference Consortium (*Church et al., 2011*) and GENCODE website (*Harrow et al., 2012*), respectively.

## Pairs of nested contact clusters

PHRIC pipeline starts with the output files of the `RNAcontacts`, which contain the coordinates of ligation junctions extracted from RIC-seq read alignments. These input files are combined into one table with an additional field listing the experiment identifier. Next, the intrachromosomal junctions below 200 nts and junctions between different chromosomes are removed, and the remaining junctions are aggregated into contacts by computing the number of supporting reads in each experiment. To cluster closely located contacts, we first cluster their left and right split points using `bedtools cluster` program (*Quinlan, 2014*), which merges split points into one cluster if the distance between them is not greater than 10 nts, as in *Margasyuk et al. (2023d)*. The contacts are combined into a cluster if their left split points belong to the same cluster, and the right split points belong to the same cluster. Each contact cluster is characterized by the number of supporting experiments and by the total number of supporting reads. Contact clusters are assigned to the positive or negative strand based on the orientation of transcripts, to which they belong.

To identify pairs of nested contact clusters (PNCC) we devised an efficient procedure based on `bedtools intersect` program (*Quinlan, 2014*). First, we identify two lists of pairs of contact clusters *A* and *B* such that (1) the left segment of *B* belongs to the window $[10; 100]$ nts from the left segment of *A*, and (2) the right segment of *B* belongs to the window $[-100; -10]$ from right segment of *A*. These lists are intersected using `bedtools intersect` as follows. We create two files in BED format that contain the coordinates of the left and the right segments of each contact cluster, respectively. Similarly, we create two additional files in BED format that store the coordinates of the left and the right segments with indents $[10; 100]$ and $[-100; -10]$, respectively. Pairwise intersection of these files and additional matching the identifiers of the constituent contacts yield the desired list of PNCC. This procedure is illustrated in Fig. S1.

The list of PNCC is filtered according to the following conditions: (1) read support of both clusters is $\geq 3$; (2) distance between clusters is between 10 and 50 nts; and (3) none of the folding intervals (intervals 1–2 and 3–4 in Fig. 1) intersects with genomic repeats annotated within RepeatMasker track of UCSC Genome Browser including SINE (short interspersed nuclear elements) and LINE (long interspersed nuclear elements), and other repeat types (*Jurka, 2000*). The condition (1) roughly corresponds to the significance cutoff for PCCRs by the frequency of forked eCLIP peaks (*Margasyuk et al., 2023a*); the condition (2) represents the range of lengths of PCCRs (*Kalmykova et al., 2021*); the condition (3) was introduced to filter out abundant RNA structures formed by low complexity regions as in previous works (*Kalmykova et al., 2021*; *Pervouchine, 2014*). The nucleotide sequences of the folding intervals are extracted from the genome using `bedtools getfasta` program (*Quinlan, 2014*).

## RNA structure prediction and classification

The secondary structure between folding intervals were predicted for each PNCC by calling the PREPH program as `fold.py -k 3 -a 3 -e -1 -u False -d 2`. That is, RNA structure was required to have a minimal helix length of 3 nts and at most two GT base

pairs in each such helix, similar settings that were used earlier to predict PCCRs (*Kalmykova et al., 2021*). There was no default threshold on the free energy and no prediction of suboptimal structures. The results of PREPH were parsed by custom scripts to extract the coordinates of the complementary regions (called left and right "handles"), their base-pairing scheme in dot-bracket notation, and the free energy of hybridization ($\Delta G$). The predicted RNA structure was categorized as conserved if both complementary parts were located within the set of conserved RNA elements (phastConsElements track for the alignment of 100 vertebrates genomes to the human genome (*Siepel et al., 2005*)), or non-conserved otherwise.

## Compensatory substitutions

To assess the abundance and statistical significance of compensatory substitutions for PHRIC predictions, including those outside of conserved 100-vertebrate alignment blocks, we employed the procedure that was outlined previously (*Kalmykova et al., 2021*). Multiple sequence alignments (MSA) of 46 mammalian genomes were downloaded from the UCSC Genome Browser website in MAF format (*Kent et al., 2002*). We analyzed pairs of alignment blocks that were cut out from the MSA by the predicted complementary regions. MSA columns containing more than 80% gaps and rows containing more than 10% gaps were removed. The resulting MSA pairs were passed to the R-scape v1.2.340 software as explained earlier (*Kalmykova et al., 2021*) after merging through a spacer containing 10 adenine nucleotides along with the restricted phylogenetic tree. The *E*-value of the structure was calculated as the product of *E*-values of its constituent base pairs that were reported by R-scape.

## PHRIC pipeline

PHRIC is implemented in a reproducible and scalable workflow management system `Snakemake`. It is available publicly through the GitHub repository (*Margasyuk et al., 2023b*).

# RESULTS

## Pairs of nested contact clusters

RIC-seq experiments in seven human cell lines (see Methods) containing 55–170 million raw reads per replicate were analyzed to extract split reads encoding RNA contacts (see Methods). In total, 2–10 million split positions supported by 15–40 million individual split reads were obtained per cell line (Fig. S2). Further analysis was confined to intragenic split reads with distance between split positions of at least 200 nts. Split points from all RIC-seq experiments were pooled and clustered using single-linkage clustering with the distance threshold of 10 nts, resulting in ~35 millions of RNA contact clusters. Due to inherent sparsity of RIC-seq data, we chose to merge split points from all RIC-seq experiments rather than to analyze them separately (*Cao et al., 2021*; *Cai et al., 2020*). Consequently, the variation in the number of split reads across cell lines did not affect the downstream analysis. Each cluster of RNA contacts was characterized by the set of RIC-seq experiments, in which it was supported, and by the total number of supporting split reads

Peer**J**

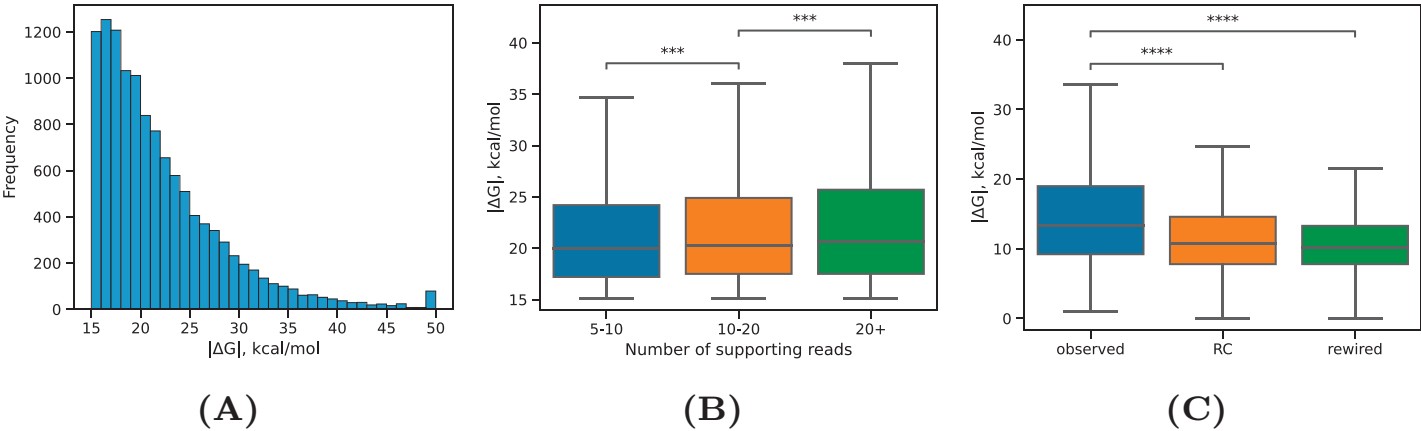

**Figure 2  RNA structure properties.** (A) The distribution of equilibrium free energies ΔG. (B) The median equilibrium free energy of RNA structure weakly increases with increasing the read support of PNCC. Sample sizes are listed in Table 1. (C) The observed RNA structures have on average larger equilibrium free energies (by absolute value) than RNA structures in the rewired set (rewired) or RNA structures formed between the interval 1–2 and the reverse complement of 3–4 (RC). Statistically discernible differences at the 0.1% and 0.01% significance level are denoted by *** and ****, respectively (two-tailed Mann-Whitney test).                              

from all RIC-seq experiments. RNA contact clusters were observed in on average 1.34 RIC-seq experiments and were supported by on average 2.05 split reads. Then, we found PNCC with distances between contacts ranging from 10 to 100 nts (Fig. 1). The resulting set of ~2.6 million PNCC was further filtered to constrain the distance between the outer and the inner contacts to be between 10 and 50 nts while requiring them to be supported by at least three split reads and excluding clusters that intersect annotated genomic repeats (see Methods). This resulted in ~29,000 PNCC, which were next passed to PREPH (*Kalmykova et al., 2021*) to predict RNA structure.

## RNA structure properties

PREPH predicts nearly-perfect stretches of complementary nucleotides in a pair of input sequences using the dynamic programming matrix based on precomputed helix energies for all *k*-mers and energies of short internal loops and bulges (*Kalmykova et al., 2021*). In application to the set of ~29,000 PNCC supported by at least three split reads, it yielded 11,998 predicted RNA structures at the equilibrium free energy cutoff $\Delta G < -15$ kcal/mol having a decaying $\Delta G$ distribution with the median of $-23.1$ kcal/mol (Fig. 2A). We subdivided the predicted RNA structures into three categories: intronic, in which both sequences were located entirely in introns; exonic, in which both sequences were located entirely in exons; and mixed, which was comprised of cases when one of the sequences overlapped a splice site or an exonic sequence contacted an intronic sequence. In all three groups, approximately 40% of the predictions were supported by 5–10 reads, roughly 40% of the predictions were supported by 10–20 reads, and 20% of the predictions were supported by more than 20 reads (Table 1). The boundaries defining these groups represent natural cutoffs subdividing the set of predictions into subsets of roughly the same cardinality with increasing read support level. Remarkably, more than 70% of the predicted structures were located outside of genomic blocks conserved in 100 vertebrates.

**Table 1 The number of RNA structures in exonic, intronic, and mixed regions by read support of PNCC.**

| Class | Total | 5–10 | 10–20 | >20 |
|---|---|---|---|---|
| Exonic | 3,676 | 1,317 (36%) | 1,634 (44%) | 725 (20%) |
| Intronic | 7,132 | 2,425 (34%) | 2,943 (41%) | 1,764 (25%) |
| Mixed | 1,190 | 489 (41%) | 499 (42%) | 202 (17%) |
| Total | 11,998 | 4,231 (35%) | 5,076 (42%) | 2,691 (22%) |

**Note:**
The percentages are with respect to the row total.

Our expectation was that PNCC with higher levels of RIC-seq support yield more stable RNA structures. Indeed, the median absolute value of $\Delta G$ increased with increasing $r$, the total number of supporting reads (Fig. 2B), but the magnitude of this increase was very small (on average, 0.03 kcal/mol with each additional supporting read). We next asked if the observed free energy distribution is non-random with respect to the rewired control (*Pervouchine et al., 2012*), in which the interacting sequences were randomly exchanged (see Methods). To control for the dinucleotide frequencies, which affect $\Delta G$ values, we separately estimated the free energy of interaction in PNCC, in which one of the nucleotide sequences was reverse complemented. Both the rewired pairs and the reverse complemented control resulted in significantly lower median $\Delta G$ values (Fig. 2C). This findings indicate that RNA structures predicted by the PHRIC pipeline are more stable than randomly occurring structures, and that their stability correlates with RIC-seq read support.

## RNA structure in exons and introns

Previous studies of RNA secondary structure and RBP interaction landscapes in eukaryotic nuclei demonstrated that introns tend to be more structured than exons (*Gosai et al., 2015*; *Sun et al., 2019*; *Zafrir & Tuller, 2015*). However, these studies assessed the propensity of RNA bases to be involved in local RNA structure and did not consider long-range RNA structure organization. We revisited this question by subdividing the exonic, intronic, and mixed into four free energy groups, 15–20, 20–25, 25–30, and >30 kcal/mol (by absolute value) as in *Kalmykova et al. (2021)* and, on one hand, into groups with high ($r \geq 12$) and low ($r < 12$) read support according to PNCC, to which they belonged. The boundary of $r = 12$ is equal to the median of the read support distribution.

The intronic RNA structures were characterized by a higher proportion of predictions with free energies exceeding 25 kcal/mol by absolute value (Table 2). Furthermore, the intronic RNA structures with high read support had a significantly larger $\Delta G$ values than those with low read support ($P$-value < 0.1%, Mann-Whitney test), while in exonic and mixed groups the difference between the high and the low read support groups was not significant (Fig. 3A). Next, we compared the $\Delta G$ values between the observed and the rewired sets of PNCC in the intronic, exonic, and other groups. The median free energies in all three groups were significantly larger as compared to the rewired control set ($P$-value < 0.1%, Mann-Whitney test), with a larger magnitude of difference for the intronic

**Table 2 The number of RNA structures in exonic, intronic, and mixed regions by free energy groups (15–20, 20–25, 25–30, and >30 kcal/mol, by absolute value).**

| Class | 15–20 | 20–25 | 25–30 | >30 |
|---|---|---|---|---|
| Exonic | 2,055 (56%) | 989 (27%) | 420 (11%) | 212 (6%) |
| Intronic | 3,189 (45%) | 1,949 (27%) | 1,042 (15%) | 952 (13%) |
| Mixed | 570 (48%) | 349 (29%) | 165 (14%) | 106 (9%) |

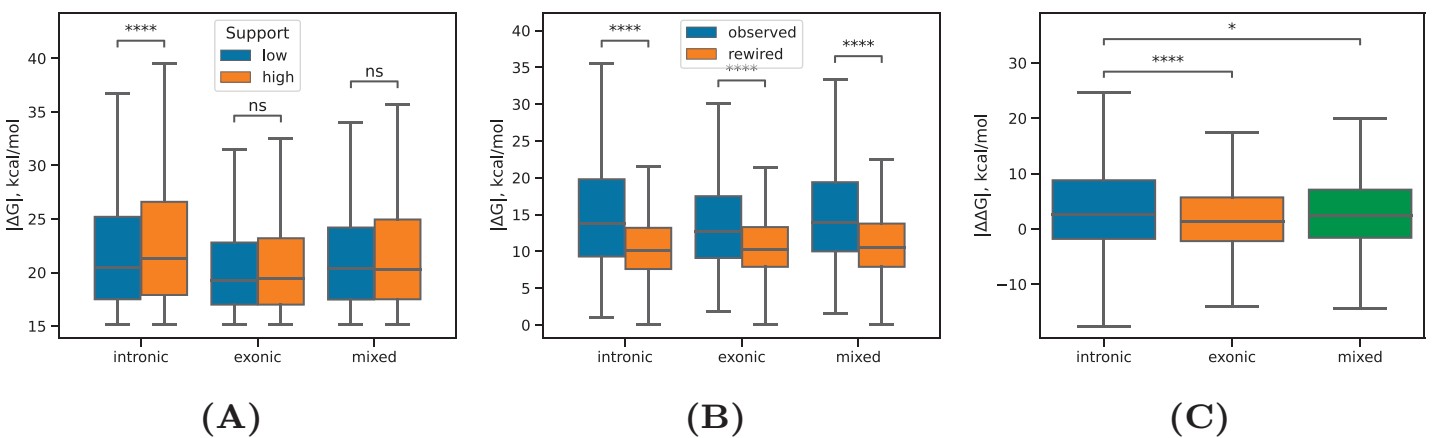

**(A)**      **(B)**      **(C)**

**Figure 3 RNA structure in exons and introns.** (A) The equilibrium free energies of RNA structure formed between intronic regions (intronic), exonic regions (exonic), and in the case when one of the sequences overlaps a splice site (mixed) for PNCC with high read support ($n \geq 12$) and PNCC with low read support ($n < 12$). (B) The equilibrium free energies of intronic, exonic, and mixed RNA structures as compared to the equilibrium free energies in the rewired control. (C) The distribution of $\Delta\Delta G = \Delta G_{obs} - \Delta G_{RC}$ values, where $\Delta G_{obs}$ is the free energy of the observed RNA structure and $\Delta G_{RC}$ is the free energy of the structure, in which one of the sequences was reverse complemented. Statistically discernible differences at the 5%, 0.01% significance level and non-significant differences are denoted by *, ****, and 'ns', respectively (two-tailed Mann-Whitney test). Sample sizes are listed in Table 1.

group than for the exonic group (Fig. 3B). Finally, we considered the distribution of $\Delta\Delta G = \Delta G_{obs} - \Delta G_{RC}$ values in the matched set of the RNA structures that were actually observed (with equilibrium free energy $\Delta G_{obs}$) and the control set, in which one of the sequences was reverse complemented (with equilibrium free energy $\Delta G_{RC}$). Again, intronic RNA structures were more stable with respect to the reverse complemented control compared to exonic and mixed groups (Fig. 3C), thus confirming that introns have a higher propensity to form stable long-range RNA structures between each other.

Next, we focused on a subset of intronic RNA structures that loop out at least one annotated exon and computed the inclusion rates $\Psi$ of these exons across all cell lines (Fig. 4A). The median $\Psi$ value decreased significantly for exons that are looped out by intronic RNA structures with increasing the read support (P-value < 1%, Mann-Whitney test), in full agreement with previous findings that exon inclusion drops with increasing stability of the surrounding RNA structure (*Kalmykova et al., 2021*; *Margasyuk et al., 2023a*), which positively correlates with the read support as we showed before (Fig. 2B).

A comparison of $\Delta G$ values between RNA structures located in conserved and non-conserved parts of exonic and intronic regions revealed that non-conserved intronic

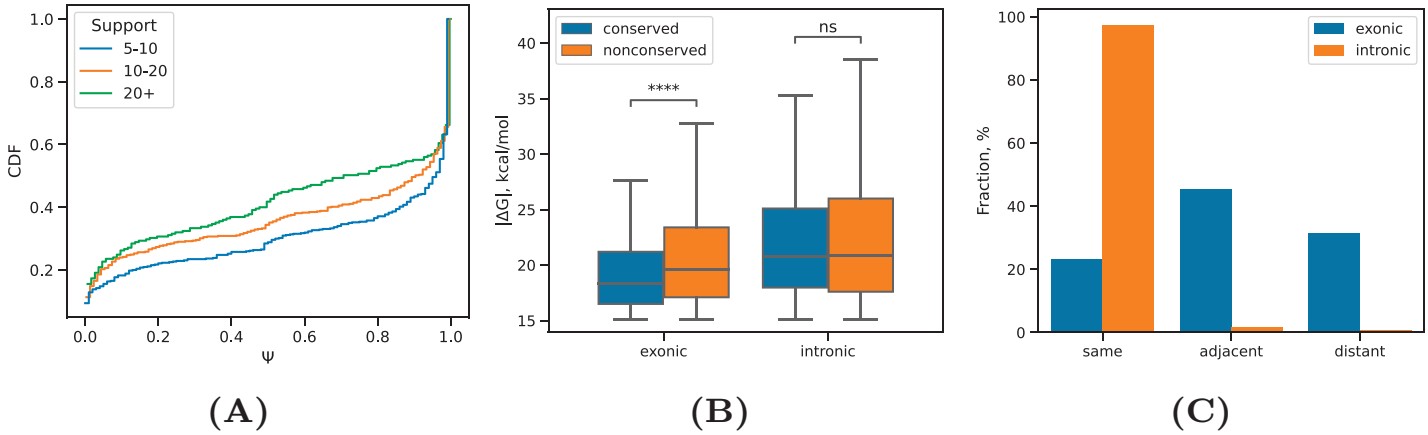

**Figure 4 RNA structure in exons and introns.** (A) The distribution of inclusion rates (Ψ) of exons looped out by the predicted RNA structures at increasing RIC-seq read support. (B) The equilibrium free energies of exonic and intronic RNA structures located in conserved and non-conserved regions. (C) The proportion of intronic (exonic) RNA structures within the same intron (exon), adjacent, *i.e.*, consecutive introns (exons), and distant, *i.e.*, non-consecutive introns (exons). Statistically discernible differences at the 0.01% significance level and non-significant differences are denoted by **** and 'ns', respectively (two-tailed Mann-Whitney test).

RNA structures were as stable as conserved intronic RNA structures, while non-conserved exonic RNA structures were on average even more stable than conserved exonic RNA structures (Fig. 4B). This indicates that exonic sequences lacking constraints on maintaining the encoded aminoacid sequence can evolve more stable RNA structures.

Finally, we subdivided intronic and exonic RNA structures into three classes corresponding to interactions within the same intron or exon, adjacent (*i.e.*, consecutive) introns or exons, and distant (*i.e.*, non-consecutive) introns or exons. Exonic RNA structures distributed almost equally likely between these groups, while intronic RNA structures have a strong preference to concentrate in the same intron (Fig. 4C). The distance between complementary sequences did not confound this result as the distributions of distances were almost the same (Fig. S3). This remarkable observation now provides an experiment-derived evidence for the tendency of RNA structures to concentrate within the same intron, which was suggested earlier by several bioinformatic studies (*Kalmykova et al., 2021*; *Pervouchine et al., 2012*).

## Case studies

In this section, we discuss a few examples of long-range RNA structures from the list of 11,998 PHRIC predictions. The entire list is available as File S1, which can be visualized through UCSC genome browser along with RIC-seq contacts in File S2 (*Margasyuk et al., 2023c*).

The human *GANAB* gene encodes the glucosidase IIα subunit and is involved in autosomal-dominant polycystic kidney and liver disease (*Porath et al., 2016*; *Besse et al., 2018*). One of its internal exons, exon 6, is spliced alternatively. We detected two PNCC surrounding exon 6 that are supported by a large number of RIC-seq split reads (Fig. 5A). PHRIC pipeline predicts two pairs of complementary regions that constitute long-range RNA structures with free energies of −26.3 and −22.1 kcal/mol, respectively. We

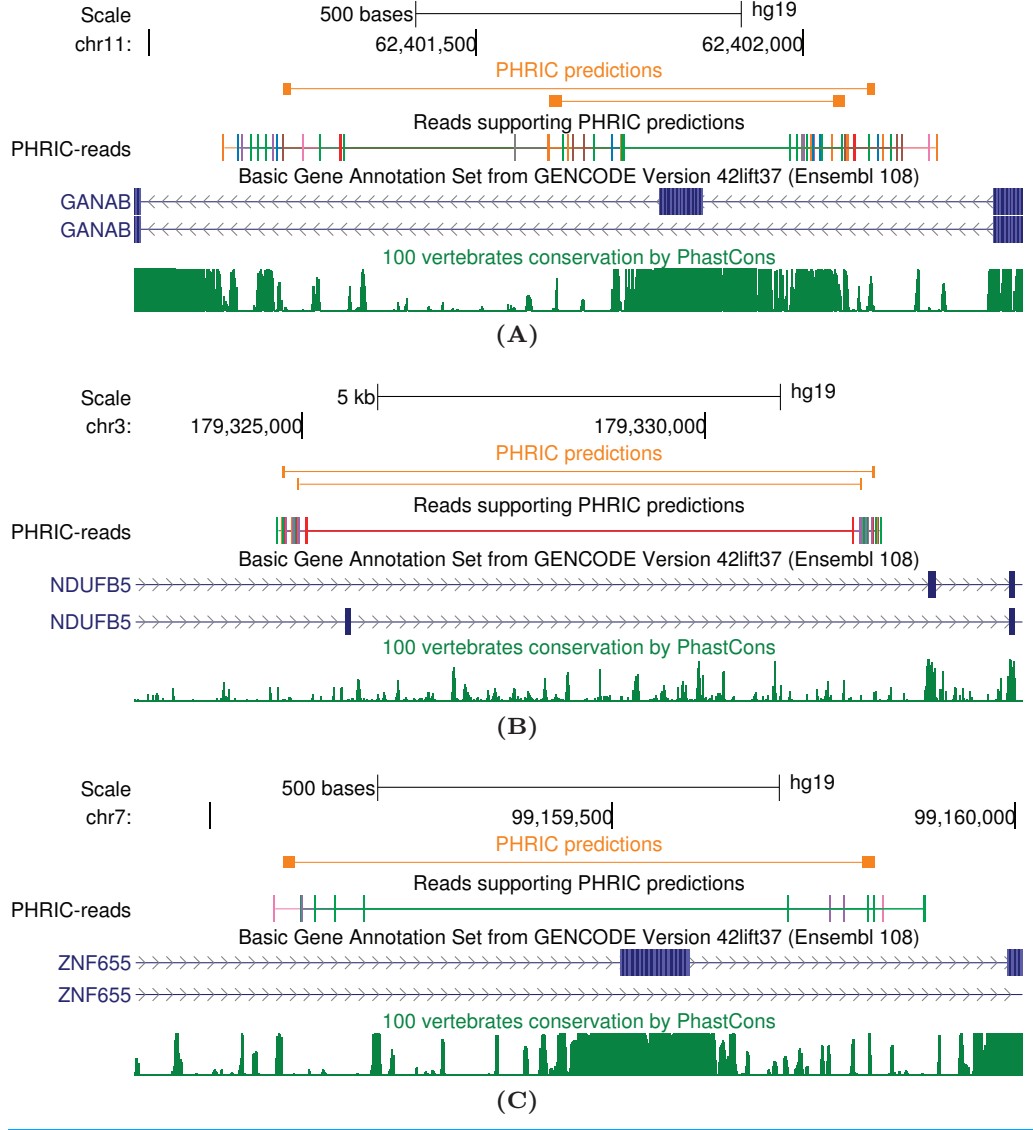

**Figure 5 Case studies of long-range intronic RNA structures in the human transcriptome outside of conserved regions.** (A) *GANAB*, the glucosidase IIα subunit. (B) *NDUFB5*, a subunit of the multisubunit NADH-ubiquinone oxidoreductase. (C) *ZNF655*, a zinc finger protein possibly involved in transcription regulation. PHRIC predictions (orange) supported by RIC-seq reads don't overlap peaks in 100-vertebrate conservation track (dark green).

hypothesize that complementary base pairing mediated by these RNA structures are responsible for alternative splicing of exon 6.

Another example of PHRIC predictions is the RNA structure in the human *NDUFB5* gene, which encodes a subunit of the multisubunit NADH-ubiquinone oxidoreductase (complex I). Three transcript variants encoding different splice isoforms have been found for this gene, and two of them differ by alternative inclusion of exon 7 (Fig. 5B). We detected two pairs of complementary intronic sequences that are supported by clusters of RIC-seq contacts. Mutually exclusive splicing of exon 7 could be modulated by these RNA structures with free energies −28.8 and −22.1 kcal/mol.

Finally, we discuss the *ZNF655* gene, which encodes a zinc finger protein that is possibly involved in transcriptional regulation. It accelerates the progression of pancreatic cancer by promoting the binding of E2F1 and CDK1 (*Shao et al., 2022*). It contains a cassette exon 4, which is surrounded by a pair of complementary sequences capable forming a duplex with free energy −25.9 kcal/mol. Alternative splicing of this exon could also be modulated by RNA structure.

## Evolutionary signatures beyond sequence conservation in vertebrates

Our earlier study has identified pairs of conserved complementary regions (*Kalmykova et al., 2021*) within the so-called conserved RNA elements, which are derived from `Multiz` sequence alignments of 100 vertebrate genomes using phylo-HMM (*Felsenstein & Churchill, 1996*; *Blanchette et al., 2004*). The generative model of phylo-HMM contains a state for conserved sites and a state for non-conserved sites, transitions between which determine the borders of conserved RNA elements. These borders and the set of conserved RNA elements itself vary depending on the set of species that were passed to the model as an input.

In this section, we asked whether the evolutionary signatures of RNA structure can be extracted directly from genomic multiple sequence alignments. Towards this goal, we restricted our analysis to 46 mammalian genomes including the human genome (*Blanchette et al., 2004*; *Kent et al., 2002*), and applied the R-scape program (*Rivas, Clements & Eddy, 2017*), which scores independent occurrence of compensatory substitutions on different branches of the phylogenetic tree, to the alignment blocks cut out by PHRIC predictions. The statistical significance of pairwise covariations in each predicted structure was estimated as a product of *E*-values reported by R-scape for all its base pairs. Out of 11,998 pairs of complementary regions originally predicted by PHRIC, 11,224 had at least one base pair with *E*-value < 1, and only 308 pairs had a significant *E*-value (below 5%) after Benjamini-Hochberg adjustment for multiple testing.

As in *Kalmykova et al. (2021)*, we found that RNA structures with significant compensatory substitutions had significantly larger equilibrium free energies (by absolute value) than RNA structures with non-significant covariations (*P*-value < 0.001, Mann-Whitney test) (Fig. 6A). The median lengths of the base-paired regions did not differ significantly between these two sets (*P*-value = 0.15, Mann-Whitney test), thus excluding the possibility that longer RNA structures contribute simultaneously to both $\Delta G$ and the *E*-value.

An RNA structure with many independent compensatory substitutions (*E*-value = $1.6 \cdot 10^{-14}$) was detected in *NFAT5* gene, a member of the nuclear factors of activated T cells family of transcription factors, which plays a central role in inducible gene transcription during the immune response (*Neuhofer, 2010*). The intron spanning between exons 7 and 8 of *NFAT5* contains a pair of complementary sequences, R1 and R2, which are strongly supported by RIC-seq as PNCC, but fall outside of conserved RNA elements in vertebrates (Fig. 6B). Manual inspection of the multiple sequence alignment of mammalian homologs revealed that intronic sequences corresponding to R1 and R2 are missing in rodents, but present in primates, canids, and African elephant, with multiple

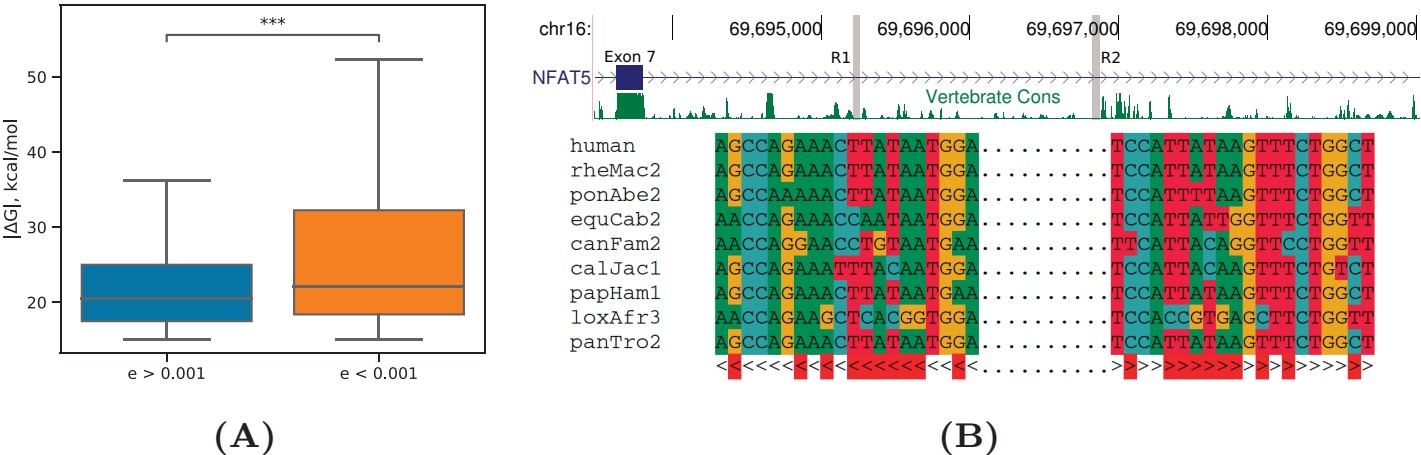

**Figure 6 Evolutionary signatures of PHRIC predictions.** (A) Equilibrium free energies ($\Delta G$) are significantly larger by absolute value for RNA structures with significant compensatory substitutions ($E < 0.001$) than for other RNA structures ($E \geq 0.001$). Statistically discernible differences at the 0.1% significance level are denoted by *** (one-tailed Mann-Whitney test). (B) A fragment of the human *NFAT5* gene between exons 7 and 8; R1 and R2 denote the complementary sequences predicted by PHRIC, which fall outside of conserved RNA elements in vertebrates (top). Multiple sequence alignment of R1 and R2 and the consensus RNA structure (bottom). Compensatory substitutions are marked in red.

independent compensatory substitutions. Remarkably, the structure formed by R1 and R2 is nested within a larger conserved structure (id355386, $\Delta G = -40.1$ kcal/mol) that was predicted for this gene earlier (*Kalmykova et al., 2021*).

## DISCUSSION

Previous studies based on RNA secondary structure profiling demonstrated that introns are generally more structured than exons (*Gosai et al., 2015*; *Sun et al., 2019*; *Zafrir & Tuller, 2015*). However, RNA secondary structure profiling can only tell whether an RNA base is paired, but it cannot tell to which other base. Here, we substantially extended this result by showing that introns have a higher propensity to form stable long-range RNA structures between each other, as compared to exons. Furthermore, we demonstrated that intronic RNA structures tend to cluster within the same intron, whereas exonic RNA structures are formed almost equally likely between consecutive and distant exons, in accordance with the knowledge that spliced mRNAs are relatively unstructured, presumably because of unwinding by the ribosome (*Rouskin et al., 2014*). These results for the first time provide experimental evidence for the tendency of RNA structures to concentrate within the same intron, thus possibly carrying information on the "splicing code".

Various high-throughput studies proposed a regulatory role of RNA structure in the control of alternative splicing (*Lu et al., 2016*; *Aw et al., 2016*; *Cai et al., 2020*). On the mechanistic level, this regulation may operate through the formation of long-range RNA structure around cassette exons promoting their skipping by the looping-out mechanism (*Nasim et al., 2002*; *Tang et al., 2020*). Computational studies based on evolutionary conservation alone and in combination with the evidence from proximity ligation experiments confirmed widespread occurrence of this mechanism (*Kalmykova et al., 2021*;

*Margasyuk et al., 2023a*), while the results obtained here extend this observation beyond evolutionarily conserved regions. The looping-out mechanism assumes bridging distant cis-elements by long-range RNA structure, which facilitates intron definition resulting in exon skipping. As we observed here, the same logic applies not only to alternative but also to constitutive splicing events since RNA structures prefer to reside within the same intron, again extending the observation made previously for PCCRs to non-conserved regions (*Kalmykova et al., 2021*). Remarkably, only a small fraction of RNA structures (below 3%) span adjacent introns, which roughly corresponds to the proportion of alternative splicing events that are actually expressed in human cell lines.

A proxy for biological function is evolutionary conservation, which in the case of RNA structure correlates with the free energy of formation. Therefore, one would expect that conserved RNA structures would be more thermodynamically stable than non-conserved structures. We demonstrated here that this is not true: introns of human genes contain almost 20 times more RNA structures in non-conserved regions as compared to conserved regions, and yet RNA structures in non-conserved regions are at least as stable as those in conserved ones. The notion of conserved regions, however, is relative to the group of species being considered. It was demonstrated earlier that pairs of complementary regions that are conserved among vertebrates are strongly supported by RIC-seq (*Margasyuk et al., 2023a*; *Kalmykova et al., 2021*). In this study, most of the predicted base pairings fall beyond vertebrate conserved regions, yet some of them show a remarkable pattern of compensatory substitutions when a smaller set of species is considered. Thus, the central question in identifying functional RNA structures beyond phylo-HMM predictions is how to combine the phylogenetic information, complementarity, and experimental evidence from proximity ligation assays into one model that accurately scores compensatory substitutions. The approach taken here disregards phylogenetic signatures instead focusing on the stable part of the RNA structure and its RIC-seq support. Future studies aimed at predicting global RNA structure, possibly including interactions with RNA-binding proteins, will have to address these concerns (*Pervouchine, 2018*).

RNA *in situ* conformation sequencing technology currently is in its infancy, but its capabilities greatly exceed those of other similar methods including PARIS (*Lu et al., 2016*), LIGR-seq (*Sharma et al., 2016*), SPLASH (*Aw et al., 2016*), and COMRADES (*Ziv et al., 2018*). In this work, we chose to pool several RIC-seq experiments together because each individual experiment yields sparse RNA contacts. Future studies, in which many more similar datasets will become available, will allow a better evaluation of the statistical significance of these contacts. The methodology developed here in the PHRIC pipeline (Fig. 1) remains applicable to the identification of locally-stable RNA structures that are supported by bilateral contacts observed in RNA proximity ligation assays.

## CONCLUSIONS

We presented PHRIC, a pipeline for identifying core elements of long-range RNA structure using RNA *in situ* conformation sequencing (RIC-seq), applied it to RIC-seq experiments in eight human cell lines, and obtained a list of ~12,000 RNA structures, most

of which belong to non-conserved regions. Our results for the first time extend RNA structure prediction in human genes beyond conserved sequence blocks.

### Funding
This work was supported by the research grant of Russian Ministry of Science and Education (075-10-2021-116) and the research grant from the National Key Research and Development Program of China (2021YFE0114900). The funders had no role in study design, data collection and analysis, decision to publish, or preparation of the manuscript.

### Grant Disclosures
The following grant information was disclosed by the authors:
Russian Ministry of Science and Education: 075-10-2021-116.
National Key Research and Development Program of China: 2021YFE0114900.

### Competing Interests
The authors declare that they have no competing interests.

### Author Contributions
- Sergey Margasyuk conceived and designed the experiments, performed the experiments, analyzed the data, prepared figures and/or tables, authored or reviewed drafts of the article, and approved the final draft.
- Lev Zavileyskiy performed the experiments, analyzed the data, prepared figures and/or tables, authored or reviewed drafts of the article, and approved the final draft.
- Changchang Cao performed the experiments, analyzed the data, authored or reviewed drafts of the article, provided early access to RIC-seq data, and approved the final draft.
- Dmitri Pervouchine conceived and designed the experiments, analyzed the data, prepared figures and/or tables, authored or reviewed drafts of the article, and approved the final draft.

### Data Availability
The prediction of complementary regions between RNA-RNA contacts derived from RIC-seq data is available at Zenodo: Sergey Margasyuk. (2023). smargasyuk/PHRIC: v0.1.0 (v0.1.0). Zenodo. https://doi.org/10.5281/zenodo.8387649.
The supplemental data is available at Zenodo: Sergey Margasyuk, Lev Zavileisky, Changchang Cao, & Dmitry D. Pervouchine. (2023). smargasyuk/PHRIC-supplementary: v0.1.0 (v0.1.0) [Data set]. Zenodo. https://doi.org/10.5281/zenodo.8199137.

### Supplemental Information
Supplemental information for this article can be found online at http://dx.doi.org/10.7717/peerj.16414#supplemental-information.

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
