# Peer review of "Long-range RNA structures in the human transcriptome beyond evolutionarily conserved regions"

_PeerJ, doi:10.7717/peerj.16414_

## Round 0.1 · original submission · Major Revisions

Dear Dr. Margasyuk and colleagues:

Thanks for submitting your manuscript to PeerJ. I have now received two independent reviews of your work, and as you will see, the reviewers raised some minor concerns about the research (mostly the manuscript format and content). Despite this, these reviewers are optimistic about your work and the potential impact it will have on research studying detection of novel and unappreciated long-range RNA structures. Thus, I encourage you to revise your manuscript, accordingly, considering all the concerns raised by both reviewers.

While the concerns of the reviewers are relatively minor, this is a major revision to ensure that the original reviewers have a chance to evaluate your responses to their concerns. There are not too many suggestions; thus, it should not take much effort to address these concerns to greatly improve your manuscript.

Please ensure that the figures are clear and that your code and related methods are easy to use with overall workflow(s) repeatable.

I look forward to seeing your revision, and thanks again for submitting your work to PeerJ.

Good luck with your revision,

-joe

Reviewer 1 ·

Basic reporting

I was able to download and run the PHRIC code. However, it was challenging as it still relies on the outdated version 2 series of Python. Also, PHRIC is available from GitHub; it must be submitted to an immutable repository such as Zenodo before acceptance.

Experimental design

Figure 2B, 2C, 3A, 3B, 3C, 4B, and 6A are problematic. It is unclear what the extent of the boxes in these box plots correspond to (standard deviation or some percentage of the data?). The number of data (N = ??) for each box is missing. It is then unclear how to interpret the significance values. If the amount of data is very large, then any small difference between the categories will be statistically significant. For example in Figure 2B the difference between the three categories is minimal, and it is surprising that these differences are statistically significant. Is this due to the large amount of data in each category?

Validity of the findings

The last line of the Abstract (“These results for the first time extend RNA structure prediction in human genes beyond conserved regions”) is an overstatement; for example, well-established programs such as ViennaRNA can do this.

Reviewer 2 ·

Basic reporting

This manuscript extends current knowledge and provides a significant novel insight that introns have a higher propensity to form stable long-range RNA structures between each other, and these RNA structures tend to cluster within the same intron. The paper is well written, the figures are detailed & relevant, and the data & code are provided for reproducibility.

Experimental design

The methods section is comprehensive and provides a detailed account of the experimental procedures and data analysis. I suggest that sharing the reasoning for certain choices (such as the parameters for data processing steps and thresholds) would improve readability and usefulness to a broader audience.

Suggestions & questions:
The section on data processing steps is quite detailed, which is good for reproducibility. However, it might be helpful for a potential reader if you could provide a schematic diagram showing the main steps.

When discussing the criteria for filtering contact clusters (e.g., read support, distance between clusters, exclusion of genomic repeats), it would be valuable to provide a brief justification for these specific criteria. Why were these thresholds chosen, and what do they imply in terms of biological significance? Additionally, were these the same thresholds used in the config.yaml distributed with the latest PHRIC code?

In the section on RNA structure prediction and classification, the parameters used for the PREPH program are described. It would be helpful to briefly explain the rationale behind these parameter choices and why they are suitable for this analysis.

Fig S1 indicates difference in results between cell lines. How are these differences addressed in the analysis?

Validity of the findings

The findings are novel & substantive and build on previous work. While the results section presents a detailed analysis of the data, there might be room for further discussion of the biological implications. Expecially with regard to the main findings: introns having a higher propensity to form stable long-range RNA structures between each other and intronic RNA structures tending to concentrate within the same intron -- the authors suggest that this might carry information on the splicing code, but how might these intronic RNA structures function, and what roles could they play in gene regulation or other cellular processes? While I understand that a detailed discussion might be prohibitively long and somewhat speculative, I encourage the authors to expand on their current Discussion section and share further interpretation of the results.

Additional comments

I commend the authors for the transparency and ease-of-access of the PHRIC code on github. It was very useful to have a test dataset provided by the authors to run the code on. I have two minor observations/questions:

1. On running the test, it appears that the files results/test_hg19/S16.tsv and resources/test_results/test_hg19/S16.tsv show different results. Is this as intended? If so, please communicate this expectation to the end user in the instructions, since it was unclear if the code successfully ran to completion and generated the expected result.

2. My attempts to re-run the test code by reducing the number of cores gave the following error:

ValueError in file PHRIC/workflow/rules/test.smk, line 12.
File "PHRIC/workflow/rules/test.smk", line 12, in __rule_test
File "miniconda3/envs/snakemake/lib/python3.11/concurrent/futures/thread.py", line 58, in run
Shutting down, this might take some time.
Exiting because a job execution failed. Look above for error message

I suspect this was because the output is hard-coded to write to results/test_hg19. On deleting this folder, I was able to successfully re-run the test to completion. Perhaps this is worth noting somewhere in the README or in the config file, since a potential end user might want to quickly re-run the code with different parameters.

---

## Round 0.2 · accepted · Accept

Dear Dr. Margasyuk and colleagues:

Thanks for revising your manuscript based on the concerns raised by the reviewers. I now believe that your manuscript is suitable for publication. Congratulations! I look forward to seeing this work in print, and I anticipate it being an important resource for researchers studying detection of novel and unappreciated long-range RNA structures. Thanks again for choosing PeerJ to publish such important work.

Best,

-joe

Reviewer 1 ·

Basic reporting

no comment

Experimental design

no comment

Validity of the findings

no comment